# Part321: Recognizing 3D Object Parts from a 2D Image Using 1-Shot Annotations

## Abstract

Recognizing object parts from images plays a pivotal role in various real-world applications. However, existing work mostly learn models from large-scale 2D part annotations. In this paper, we propose a part recognition model that can recognize 3D parts from a 2D image with only annotations of parts on one 3D mesh model for each object category. Specifically, we build a category-level 3D feature bank for meshes that could overcome geometric variance among objects and precisely align with diverse 2D images of this object category. To achieve this, we propose to learn two types of correspondence. Firstly, we learn mesh-to-mesh correspondence between distinct 3D mesh models by matching geometry-aware features, which allows us to create a shared 3D feature bank for this object category. Secondly, we establish mesh-to-image correspondence by aligning features in the 3D feature bank with features extracted from 2D images. During inference, given a single image, our method recognizes 3D object parts via a Render-and-Compare approach. It predicts object parts by gradient-based optimizing each part's 3D configuration, minimizing a feature-level reconstruction loss between the projected 3D features and the image features while ensuring geometric consistency between object parts. The position, rotation, and shape of each part are optimized to match the cues from the image, thus recognizing the 3D parts from a 2D image. Experiments on VehiclePart3D, PartImageNet, and UDA Part dataset show our method outperforms baselines significantly for 2D part segmentation and pioneering 3D part recognition from a single image.

## 1 Introduction

Object part recognition from images plays an important role in real-world applications, *e.g.*, autonomous driving and embodied AI. Existing work (Liu et al., 2021; Tritrong et al., 2021) achieves object part recognition by learning segmentation models from large-scale image sets with parts annotated on each 2D image pixel. Unfortunately, annotating pixel-level object parts on large-scale training sets is tedious and requires huge efforts of human work. Moreover, current part recognition approaches are specialized to identify the exact part definition from the training labels whereas different vision tasks require different object part definitions, *e.g.*, robotics needs to detect interactable parts like handles (Ainetter & Fraundorfer, 2021), and traffic cameras need to detect parts like car doors (Morris & Trivedi, 2008). The various demands of part definition further narrow the potential applications of current approaches. While defining object parts at a more fine-grained level and merging them differently in different applications is a possible solution, it makes the annotation process almost impractical.

Cognitive psychology studies suggest that humans recognize objects as a composition of simple geometric components in 3D space (Biederman, 1987). Therefore, annotating parts and building a part recognition model in 3D space may be a natural solution to the data annotating challenge. Annotated parts in 3D space are more informative since all parts are visible and avoid the ambiguity of the part annotation on 2D images. In this paper, we aim to recognize 3D object parts from a single RGB image by inferring the 3D configuration of each object part. Moreover, our model learns from a single 3D annotation for each object category, which makes the annotation process much faster compared with 2D approaches, and addresses the part definition issue by just alternating the part definition mesh under different applications without training again.

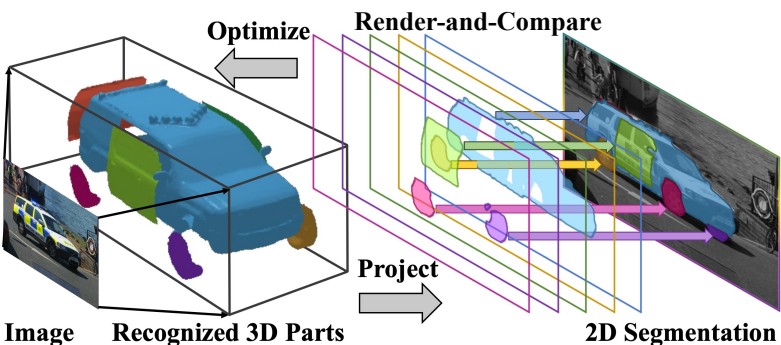

Figure 1: Part321 detects 3D object parts from a single image by Render-and-Compare correspondence features learned on the neural mesh. Through optimizing the 3D configuration of each object part, we recognize the parts in 3D and get projected 2D part segmentation.

Conducting part recognition from only a single 3D mesh annotation will introduce two challenges: (1) The geometry of different objects in one object category can vary a lot, *e.g.*, different shapes of airplanes. Thus, finding a commonly shared representation of diverse shapes is essential. (2) We need an inference algorithm, which bridges the parts on 3D meshes with the pixels on 2D images.

To solve these challenges, our idea is to form a part compositional neural mesh for the annotated 3D object, a representation in latent space that attaches features to each vertex of the mesh. The features are shared across different objects in the object category and are aligned with pixels in 2D. We further make the neural mesh deformable to overcome the shape variance among different objects. The feature sharing and deformation make our part recognition method a category-level paradigm and the 3D-to-2D alignment facilitates the render-and-compare inference algorithm on images.

To build such a neural mesh, we propose to learn two types of correspondence. First, we learn mesh-to-mesh correspondence among 3D meshes, which means for any vertex on one mesh, we can find the most geometrically similar vertex on any other mesh within the object category, thus these vertices could share the same feature in the category-level neural mesh. The correspondence is obtained by matching geometry descriptors learned in a self-supervised manner. Based on the correspondence, we further train a deformation network that could reshape the neural mesh into diverse geometries given different shape latent. Second, we establish a mesh-to-image correspondence, which aligns the 3D features on the neural mesh with the 2D features extracted from images. We first use a diffusion model prompt by meshes and camera configurations (Ma et al., 2023) to create a training set of semi-realistic images. Note that this process provides accurate ground truth poses and 3D meshes of objects in the generated images. Using the generated images, we learn the mesh-to-image correspondence by training an image feature extractor to precisely match the features on neural mesh vertices with features extracted from the pixels that the vertices are projected on.

For inference, as shown in Figure 1, given a testing image, we extract image features using the trained feature extractor and conduct feature level render-and-compare to optimize the 3D configuration of each part in the neural mesh to minimize a feature space reconstruction loss. Specifically, the 3D configuration of parts includes location, rotation, scale, and shape. For the location, rotation, and scale, we apply gradient descent on them to search for the optimal 3D pose and size. As for the shape, we optimize the shape latent, given that the trained deformation network will deform each part mesh to fit the geometries in the image. Also, the optimization involves a geometry consistency loss between object parts as additional constraints. Note this optimization process allows each part to have its own freedom for optimization while keeping the holistic spatial relations.

We evaluate our approach on both 2D and 3D part detection tasks. Our learned model can be evaluated by standard 2D object part segmentation tasks by simply projecting the reconstructed 3D parts on the image plane. We compare our approach with advanced approaches for one-shot 2D part segmentation on real-world datasets. Besides, we conduct experiments to infer 3D object parts from images and measure the accuracy. Both quantitative and qualitative results show the effectiveness of our framework in recognizing 3D parts from 2D images with only one annotation.

Our contributions can be summarized as follows:

- We propose Part321, a category-level object part recognition method that only requires a single 3D annotation, which pioneers one-shot 3D part detection from 2D images and achieves State-of-the-Art performance on one-shot 2D part segmentation.

- We propose to learn two types of correspondence to train Part321: (1) mesh-to-mesh correspondence, which establishes the category-level features among different objects and deforms part geometry using the deformation network; (2) mesh-to-image correspondence, which bridges the 2D images with the 3D meshes in feature space.

- We build a part recognition inference pipeline from 2D images using feature level render-and-compare method, where the 3D configuration (location, rotation, scale, and shape) of parts are predicted with holistic geometry constraint.

- We collect VehiclePart3D, a dataset consisting of part segmentations on 2D real images, 3D part annotations on meshes, and synthetic training images.

## 2 RELATED WORK

**Learning Object Parts in 3D.** Learning 3D parts from 3D inputs, *e.g.*, pointclouds, has been widely explored. The 3D semantic segmentation focuses on grouping 3D points into parts. Previous works have explored many effective network architectures (Qi et al., 2017a;b; Yu et al., 2019; Shi et al., 2020; Zhang et al., 2022) and training methods (Landrieu & Simonovsky, 2018; Afham et al., 2022; Liu et al., 2023; Zhang et al., 2023b) to improve the ability of models to recognize 3D parts. Another important area is the 3D part discovery, which involves decomposing 3D objects into self-defined parts. This task is critical for applications in reconstruction, assembling, and canonicalization. (Xu et al., 2019; Luo et al., 2020; Sun et al., 2021; Koo et al., 2022). However, a common limitation of these approaches is their reliance on 3D observations. As the field progresses, there is an increasing trend towards 3D object inference from single images (Wang et al., 2022). Recognizing this trend, our work intends to explore the task of detecting 3D parts from a single image.

**2D Object Part Segmentation.** Semantic object part segmentation is a long-standing problem in computer vision. The pioneering work Pictorial Structure (Fischler & Elschlager, 1973) along with following works (Weber et al., 2000; Felzenszwalb & Huttenlocher, 2005; Fei-Fei et al., 2006; Zhu & Mumford, 2007; Girshick et al., 2011) explicitly model parts and their spatial relations. These methods share a common topic that the object-part models provide rich representations of objects and help interpretability. However, in the era of deep learning with data-driven models, research on part segmentation gets hindered due to the lack of large-scale datasets. As a result, most recent works (Hung et al., 2019; Choudhury et al., 2021; Liu et al., 2021; Tritrong et al., 2021; Gadre et al., 2021; Ziegler & Asano, 2022; Saha et al., 2022) focus on unsupervised or self-supervised co-part segmentation. There are also some works (Liu et al., 2022; Peng et al., 2023) leveraging synthetic data and domain adaptations on part segmentation. In this paper, we explore a novel direction to solve 2D part segmentation by detecting the object parts in 3D space and projecting the 3D parts to generate 2D segmentations, significantly improving the data efficiency and performance.

## 3 METHOD

We formulate the object part recognition problem as establishing a part compositional neural mesh (Section 3.1) that represents the annotated object as a combination of deformable object parts with features on mesh vertices.

To achieve this, as shown in Figure 2, we build mesh-to-mesh correspondence and mesh-to-image correspondence. The mesh-to-mesh correspondence (Section 3.2) is defined as: for each vertex on one mesh, which vertices of other meshes have the highest geometric similarities. This correspondence allows us to establish the category-level shared features, which means corresponding vertices could share the same feature in the neural mesh. We also train a deformation network based on the correspondence. The mesh-to-image correspondence (Section 3.3) learns the alignment of mesh vertices with image pixels, measured by the similarities between features on the neural meshes and the features extracted from the images. We use contrastive learning to match the 3D features on vertices with their projected pixels. Such a formulation makes detecting 3D parts from 2D images approachable by differentiating the process of rendering the neural mesh into 2D feature maps.

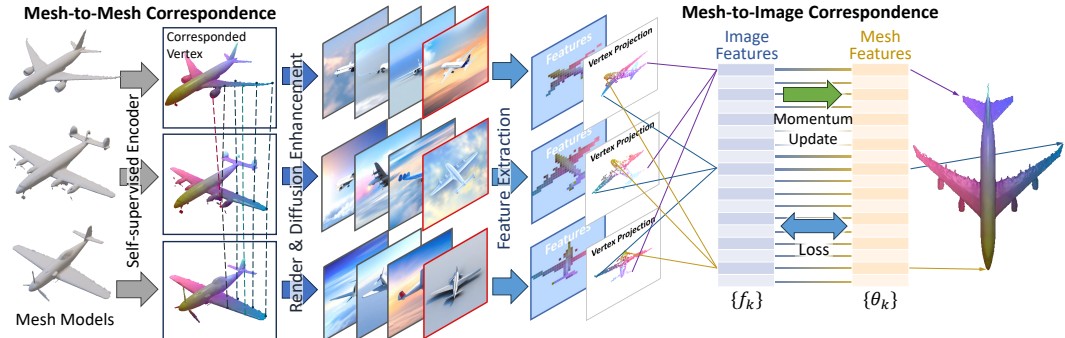

Figure 2: The overview of our training process. The mesh-to-mesh correspondence is learned on mesh vertices, which find corresponding vertex on one mesh given a vertex in another mesh. Then, semi-realistic images are generated using the DST (Ma et al., 2023). Finally, a mesh-to-image correspondence is learned to bridge the 3D meshes with the generated 2D image using feature rendering.

During inference (Section 3.4), we leverage the "render-and-compare" approach to optimize the 3D configuration of each part in the neural mesh, searching the correct position, rotation, scale, and deformation of the parts by aligning the 3D features with 2D image features.

## 3.1 PART COMPOSITIONAL NEURAL MESH

We establish part compositional neural meshes $\mathfrak{N}$, which associates the annotated meshes with geometric-aware features shared by objects within a category, and has precise alignment with 2D images. The neural mesh $\mathfrak{N} = \{\mathcal{V}, \mathcal{A}, \mathcal{U}, \mathcal{X}\}$ consists a set of mesh vertices $\mathcal{V} = \{V_k \in \mathbb{R}^3\}_{k=1}^K$, triangular faces $\mathcal{A} = \{A_k \in \mathbb{N}^3\}_{k=1}^{K'}$, feature vector on each vertex $\mathcal{U} = \{U_k \in \mathbb{R}^d\}_{k=1}^K$, and part label $\mathcal{X} = \{X_k \in \mathbb{N}\}_{k=1}^K$, where $K$ and $K'$ is the number of vertices and faces of the mesh.

In order to model the variant geometry in each object category, we introduce a deformation network $\Phi$ to reshape the geometry of the neural mesh. The location of each vertex is transformed by $V_k = \hat{V}_k + \Phi(\hat{V}_k, z_k)$, where $\hat{V}_k$ is the shape template provided by the annotated object and $z_k$ is a latent variable that controls the deformation. The learning process of $\Phi$ is described in the following section. We control the shape latent of vertices $z_k$ within each object part $\chi$ to be the same.

## 3.2 MESH-TO-MESH CORRESPONDENCE

We learn the vertex-level mesh-to-mesh correspondence between meshes $\mathfrak{N}_y$ to share features among different objects when building the neural mesh, where y is the index of mesh.

We formulate the mesh-to-mesh correspondence as feature matching, which means that we propose to learn the features on each vertex representing its geometric information and exploit the cosine similarity of feature vectors to form the correspondence. Following previous work (Sun et al., 2021), we learn a PointNet++ (Qi et al., 2017b) encoder $\Psi$ to extract object geometry descriptors unsupervisedly from pointclouds. We use the encoder to compute features for each vertex on the meshes $\gamma_{y,k} = \Psi(\hat{\mathcal{V}}_y)$, which involves a feature interpolation process (for more detail, refer to Appendix A.1). Then, we compute the cosine similarity of features to obtain the dense correspondence between vertices across all object meshes. For vertex $k_1$ on mesh $\mathfrak{N}_{y_1}$, corresponding vertex $k_2$ on mesh $\mathfrak{N}_{y_2}$ is defined as:

$$k_2 = Corr(k_1, y_2) = \mathrm{argmax}_k \frac{\gamma_{y_1,k_1} \cdot \gamma_{y_2,k}}{|\gamma_{y_1,k_1}| \, |\gamma_{y_2,k}|}. \tag{1}$$

Based on the learned correspondence, we further train the deformation network $\varphi$. For each vertex, the network takes the location of the template vertex $\{\hat{V}_{k_1}\}$ and one hot shape latent $z_y$ that represents the target mesh with index $y_2$ in the category. Using our learned mesh-to-mesh correspondence, the ground truth vertex offset can be defined as $V_{y_2, Corr(k_1, y_2)} - \hat{V}_{k_1}$. The loss for

training the part deformation network $\varphi$ is:

$$\mathcal{L}_{\text{deform}} = \sum_y \sum_k |(V_{y_2, Corr(k_1, y_2)} - \hat{V}_{k_1}) - \varphi(V_{k_1}, z_{y_2})|. \tag{2}$$

We also apply the surface-normal-consistency loss to keep the deformed mesh smooth. During inference, our framework could deform each object part into diverse shapes by changing the latent $z$. For more details about the deformation network, please refer to Appendix A.2.

### 3.3 MESH-TO-IMAGE CORRESPONDENCE

To bridge the 3D parts in the neural mesh with 2D images, we introduce the mesh-to-image correspondence, which is formulated as similarity between features on each vertex $U_k$ and the image features extracted $\Phi_w(I) = F \in \mathbb{R}^{c \times h \times w}$ from image $I$, where $\Phi$ is the feature extractor with network parameters $w$.

As shown in Figure 2, we use the semi-realistic image generated by the DST (Ma et al., 2023) for training, where the meshes set $\{\mathfrak{N}_y\}$ is rendered into synthetic images with Blender (Community, 2018) and enhanced by a ControlNet (Zhang et al., 2023a). The image generation process ensures the accurate alignment between meshes and images (*i.e.*, , camera pose and shape), and providing promising realism. Note that this training data generation process is part-agnostic, which requires no part annotation.

To learn the correspondence, we first determine the mesh $\mathfrak{N}_y$ used to generate the image. We then calculate the world-to-screen transformation $\Omega$ using the known camera pose $Q \in \mathbb{R}^3$. To find the vertex $k$ corresponding feature $f_k = F(p_k)$ at pixel $p_k$ on the feature map, we compute the projected location of each vertex on the feature map $p_k = \Omega(V_k)$. Besides, the visibility $o_k$ is determined for each vertex in the image, *i.e.*, $o_k = 1$ if vertex $k$ is visible, and vice versa. Please refer to Appendix A.3 for more details about visibility.

Our learning optimal of the 2D feature extractor is to enlarge the feature distance $\|\theta_j - f_i\|_2$ if $\|V_i - V_j\|_2$ is above a desired threshold. Such properties of features allow us to use differentiable rendering to find the optimal alignment of the vertices on the 3D model and corresponding locations on the 2D image. To achieve this, we use the contrastive loss (Bai et al., 2023; Wang et al., 2020) to learn the weights $w$:

$$\mathcal{L}_{\text{train}} = -\sum_k o_k \cdot \log(\frac{e^{\kappa f_k \cdot \theta_k}}{\sum_{\theta_l \in \mathcal{Y}_y, v_l \notin \mathcal{N}_k} e^{\kappa f_k \cdot \theta_l}}), \tag{3}$$

where $\kappa$ is a preset softmax temperature, and $\mathcal{N}_k$ indicates a spacial neighborhood of $V_k$, which controls specially accuracy of the learned features. In practice, we also include the background term to further improve the overall performance. At the same time, we adapt the vertex features $\theta_k$ using the momentum update strategy (Bai et al., 2023),

$$\theta_k \leftarrow o_k(1 - \beta) \cdot f_k + (1 - o_k + \beta \cdot o_k)\theta_k, \tag{4}$$

where $\beta$ is the momentum for the update process.

Notably, in previous object pose estimation approaches using contrastive learning (Ma et al., 2022; Wang et al., 2020; 2024), the vertex features are learned fuzzily with a prototype geometry (*e.g.*, a cuboid). In our approach, the correspondence between meshes allows us to utilize the detailed object geometry, making it feasible to learn precise descriptors $\theta_k$ of part-level structures, *e.g.*, center of the left front wheel. Such difference allows Part321 to accurately locate object parts in images.

### 3.4 PART INFERENCE

Our inference pipeline (Figure 3) predicts 3D object parts by optimizing the overall object pose and 3D configurations (*i.e.*, location, rotation, scale, and shape) of the each part in neural mesh through the feature-level rendering and comparison.

Specifically, we extract a feature map using the trained feature extractor from the testing image $F = \Phi_W(I)$. We also render a feature map $\hat{F}$ using the learned neural meshes $\mathfrak{P}_{y, \chi}$ given a set of 3D attributes, *e.g.*, shape, pose, and scale. By comparison between the two feature maps, we find

Figure 3: The inference process of Part321 . We use the deformation network to reshape the mesh given the shape latent (shown as grids with grayscale representing numerical values). Image features are extracted from the given test image. We optimize the whole object pose and 3D configuration (location, rotation, scale, and shape) of object parts by gradient-based minimizing the feature reconstruction loss. The 2D part segmentation is computed by a projection of optimized 3D parts.

cues to update the 3D attributes to make the rendered feature map better align with the extracted features. Technically, we use gradient optimization to iteratively minimize the feature difference loss on each pixel $p$ on the feature map:

$$\mathcal{L}_{\text{recon}} = \sum_p \|F_p - \hat{F}_p\|_2^2. \tag{5}$$

In detail, we first use the whole neural mesh to optimize the camera pose $R'$, which gives the initial prediction of the 3D rotation $R$ of the whole object. Then, for each object part $\chi$, we introduce additional scaling parameter $S \in \mathbb{R}^3$, shape latent $Z \in \mathbb{R}^Y$, where $Y$ is the length of the shape latent, and transformation $T \in \mathbb{R}^6$, which includes the 3D translation and 3D rotation. By changing $T$, $S$, and $Z$, each object part can move freely in the 3D space and be deformed into diverse shapes.

Also, to ensure the geometry consistency of between object parts, we introduce a geometry consistency loss among object parts. We select the paired vertices $\{V_i, V_j\}, i \in \chi, j \in \hat{\chi}$ between object parts $\chi$ and $\hat{\chi}$, which have distances $\rho_{ij} = \|V_i - V_j\|_2$ smaller than a threshold $\tau$. A consistency loss is applied if the distance of these paired vertices exceeds the threshold during optimization:

$$\mathcal{L}_{\text{consist}} = \sum_{i,j} (\rho_{ij} - \tau)\mathbb{1}[\rho_{ij} > \tau], \tag{6}$$

where $\mathbb{1}$ is an indicator function that equals 1 if the expression is true and equals 0 if otherwise.

The overall optimization loss $\mathcal{L}_{\text{inference}}$ is the weighted sum of feature difference loss and geometry consistency loss with $w_{\text{consist}}$ as the weight. We conduct gradient optimization to find 3D configuration w.r.t $R, T, S, Z$ with minimal $\mathcal{L}_{\text{inference}}$ for 300 steps, thus recognizing object parts in 3D space. For the prediction of 2D part segmentation, we simply project each part mesh $\mathfrak{P}_\chi$ with the optimized 3D configurations onto the image plane.

## 4 EXPERIMENTS

To validate the effectiveness of our framework in recognizing 3D parts from a single image, we conduct evaluations both in 3D and 2D. The reason why we also conduct 2D experiments is to compare our method with existing baselines for 2D part segmentation, while 3D part detection from images using only one annotation lacks comparable baselines. **Note that** this is an unfair comparison since our framework performs the extra task, which is more challenging than the purely 2D task.

### 4.1 SETTINGS

Both Part321 and baselines are trained on the semi-realistic training images produced by rendering CAD models using DST (Ma et al., 2023). The object CAD models are obtained from ShapeNet (Chang et al., 2015) and Objaverse (Deitke et al., 2022). During training, we use 30 CAD models and 3000 synthesized images for each category. Please see Appendix B.3 for training details.

Table 1: Quantitative results on VehiclePart3D show that despite Part321 performs the extra 3D recognition task, it outperform all 2D baselines with large-scale pretraining and pseudo labeling.

| Segmentation mIOU ↑ | Police Car | Airliner | Bicycle | Jeep | Minibus | Mean |
|---|---|---|---|---|---|---|
| SegFormer | 39.57 | 37.25 | 23.21 | 37.02 | 32.82 | 33.97 |
| DeepLab v3+ | 44.54 | 35.71 | 23.74 | 34.47 | 31.60 | 34.01 |
| SegFormer w/ Pseudo Label | 49.96 | 40.85 | 28.65 | 38.08 | 36.08 | 38.72 |
| DeepLab v3+ w/ Pseudo Label | 51.79 | 38.48 | 28.15 | 40.99 | 33.52 | 38.59 |
| Part321 | **53.61** | **41.77** | **31.67** | **42.12** | **43.92** | **42.62** |

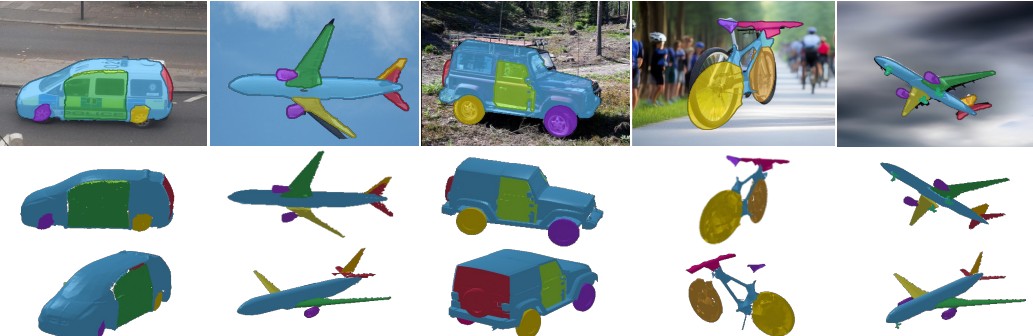

Figure 4: Qualitative results on VehiclePart3D and 3D DST. The 2D part segmentations (top) and reconstructed 3D parts in the predicted view and a novel view (middle and bottom) show that Part321 accurately recognizes object parts in both 2D and 3D. Parts are represented by different colors.

**Datasets.** We evaluate the 2D part segmentation on VehiclePart3D, PartImageNet (He et al., 2022), and UDA-Part dataset (Liu et al., 2022). The performance of 3D part detection is evaluated on the 3D-DST dataset (Ma et al., 2023). *VehiclePart3D* consists of 279 part annotated real images and 47 CAD models from 5 categories. For each category, we define 4 to 7 object parts. *PartImageNet* includes 24,000 part annotated real images from ImageNet. *UDA-Part* includes 200 annotated real images of 5 vehicle categories from PASCAL3D+ dataset (Xiang et al., 2014). *3D DST* includes 12k realistic 2D images generated from corresponding 3D models by diffusion models (Zhang et al., 2023a). We use 36 3D models from 4 categories of rigid objects. We evaluate Part321 and baselines on images generated by 5 unseen CAD models for each category. The generated images and the 3D CAD models are aligned based on the camera poses, which enables us to measure the performance of 3D part detection. Please refer to Appendix B.1 and B.2 for dataset and part annotation details.

**Baselines.** We compare our framework with SOTA 2D segmentation methods on our one-shot object part segmentation setting since there lacks one-shot 3D part detection from a single image method. The baseline approaches include SegFormer (Xie et al., 2021) and DeepLab v3+ (Chen et al., 2018). We train the baseline approaches on segmentation maps generated from the same annotated cad model as Part321. We use ResNet50 (He et al., 2016) encoder for DeepLab v3+, and MiT-B2 encoder for SegFormer. The encoders are pre-trained on ImageNet-1K dataset (Deng et al., 2009). To further enhance the baseline approaches, we apply the pseudo-labeling techniques from a SOTA domain adaptation method (Hoyer et al., 2022). The baselines always have the same amount of labeled training images as Part321. For more details about baselines, please refer to Appendix B.5.

**Metrics.** We use Mean Intersection over Union (mIoU) as the metric for the 2D part segmentation task, where IoU is first computed for each part and background, and then averaged over all classes. We use Chamfer Distance (CD) and 3D Bounding Box IOU to evaluate 3D part recognition accuracy. We evaluate the 3D parts after the camera and parts transformations. The percentage of predicted 3D pose with rotation error smaller than $\frac{\pi}{6}$ is used to evaluate the object 3D pose estimation.

### 4.2 2D OBJECT PART SEGMENTATION

Table 1 and Figure 4 show the results of one-shot object part segmentation on VehiclePart3D. The results demonstrate that Part321 outperforms all baseline approaches. The visualizations show that our 2D segmentations are obtained from an accurate estimation of the 3D geometry of the objects.

Table 2: Quantitative results of 2D segmentation on PartImageNet dataset show that Part321 outperforms baselines significantly when annotations are fine-grained (*e.g.*, Cars have 9 parts) while keeps comparable performance when coarse annotations favor 2D methods (*e.g.*, Boats have 1 or 2 parts).

| Segmentation mIOU ↑ | Car | Aeroplane | Bicycle | Boat | Mean |
|---|---|---|---|---|---|
| SegFormer | 28.85 | 39.68 | 26.72 | 47.21 | 37.47 |
| DeepLab v3+ | 28.73 | 38.69 | 34.76 | 43.02 | 37.23 |
| SegFormer w/ Pseudo Label | 26.42 | 44.05 | **45.58** | **48.66** | 42.73 |
| DeepLab v3+ w/ Pseudo Label | 30.44 | 38.10 | 45.12 | 45.03 | 42.99 |
| Part321 | **50.47** | **45.14** | 40.07 | 42.51 | **46.12** |

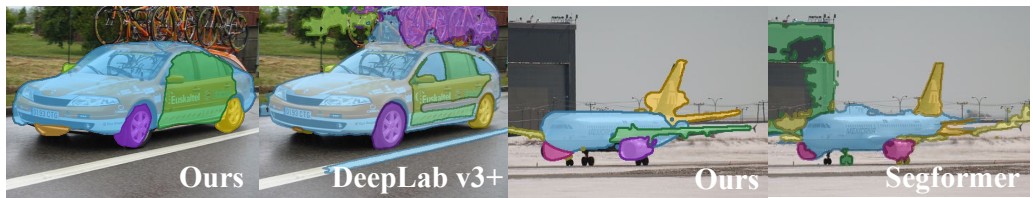

Figure 5: Qualitative results of 2D part segmentation on PartImageNet dataset show that our prediction is more geometry-aware and is less affected by the complex background. Parts are represented by different color masks with highlighted boundaries.

Table 3: Quantitative results for part segmentation on UDA Part show the robustness of our method on different dataset. *w/ Pseudo* denotes the baseline method trained with pseudo labeling techniques.

| Segmentation mIOU ↑ | Car | Aeroplane | Bicycle | Mean |
|---|---|---|---|---|
| SegFormer | 24.33 | 35.92 | 38.30 | 32.85 |
| SegFormer w/ Pseudo | 25.60 | 39.64 | **47.40** | 37.55 |
| Part321 | **37.39** | **39.88** | 44.89 | **40.72** |

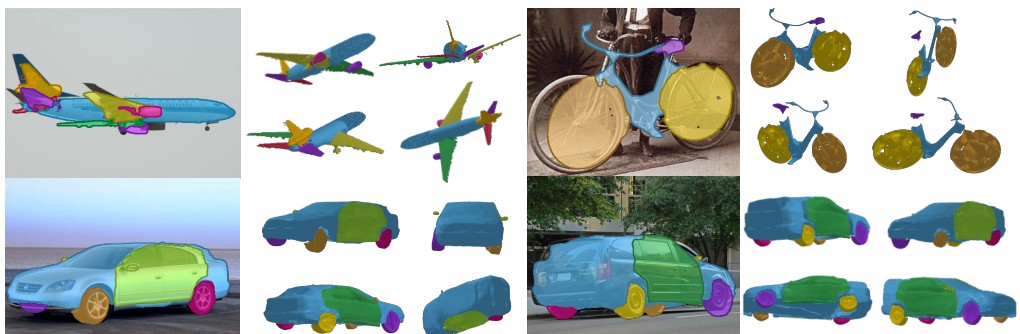

Figure 6: Qualitative results on UDA part. We visualize the 2D segmentation, where parts are represented by different colors with highlighted boundaries. 3D parts reconstructed from the images are shown on the right, which are aligned with the segmentation.

Table 2 and Figure 5 show the results on PartImageNet. The results show that Part321 has a better one-shot part segmentation ability compared with SOTA baselines. According to the visualizations, Part321 suffers less error from the background patterns and is able to accurately recognize spatial relationships (e.g., the right wing and left wing). Note for the boat category, each image only contains one or two parts, thus is more similar to an instance segmentation instead of part segmentation.

Table 3 and Figure 6 show the quantitative and qualitative comparison between Part321 with baselines on the UDA part. The results show Part321 has a higher part segmentation accuracy especially when part definitions are more fine-grained. For more visualizations, please refer to Appendix C.

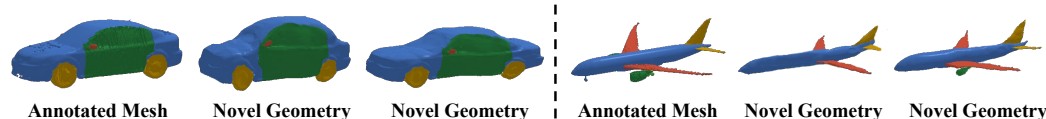

| Annotated Mesh | Novel Geometry | Novel Geometry | Annotated Mesh | Novel Geometry | Novel Geometry |

Figure 7: Qualitative results for mesh deformation network. Our network could deform the annotated mesh into diverse geometries while keeping the part annotation.

Table 4: Quantitative results for 3D part detection on 3D DST. We use the 3D pose estimation accuracy, chamfer distance and 3D Bounding Box IoU to show that our method could precisely recognize 3D object parts from a single image with accurate pose, shape and location.

|  | 3D Pose Accuracy ↑ | Chamfer Distance ↓ | 3D Bounding Box IOU ↑ |
|---|---|---|---|
| Part321 | 80.40 % | $3.019 \times 10^{-2}$ | 40.56 |

Table 5: Ablation Study on 3D DST and PartImageNet dataset. We measure 2D mIoU and 3D part detection accuracy to validate the necessity of our different components in both 2D and 3D tasks.

|  | 3D DST | | | PartImageNet |
|---|---|---|---|---|
|  | 2D mIOU ↑ | CD $(10^{-2})$ ↓ | 3D IOU ↑ | 2D mIOU ↑ |
| w/o Part Scaling | 33.22 | 3.206 | 32.43 | 42.18 |
| w/o Geo Constrain Loss | 38.47 | 3.502 | 37.54 | 44.17 |
| w/o Part Deformation | 36.91 | 3.404 | 39.57 | 43.59 |
| Full Model | **46.12** | **3.019** | **40.56** | **47.30** |

## 4.3 3D OBJECT PART DETECTION

Table 4 shows the quantitative evaluation for 3D Object Part Detection from images. The performance shows that our model could correctly locate the object parts in 3D space and deform the parts into suitable geometries. Figures 4 and 6 show the visualization of our 3D object part detection on VehiclePart3D and 3D DST datasets. The results show that the parts with optimized 3D configuration closely resemble the parts in the image.

As shown in Figure 7, using the trained deformation network, we could reshape the annotated mesh into diverse geometries with consistent part annotation, making the shape optimization possible. Specifically, we use the shape latent to control the deformation, and different latent will lead to different goal shapes. During inference, the latent will be optimized, finding the most suitable shape for each 3D part to align with the image.

## 4.4 ABLATION STUDY

Tabel 5 shows the ablation study of important components in our approach. In the *w/o Part Scaling* setup, the object scale $S$ is set to be fixed during part optimization. For *w/o Geo Constrain Loss*, we remove the geometry consistency loss during optimization. The *w/o Part Deformation* setting shows the results that no shape deformation are applied during inference. The results show that all the proposed components are essential to achieve the 3D part detection ability.

## 5 CONCLUSIONS

We present Part321 to address the challenge of recognizing 3D object parts from a 2D image using one-shot annotation. Our method establishes a **part compositional neural mesh** by introducing two correspondences: **mesh-to-mesh correspondence**, enables sharing features in the same category; and **mesh-to-image correspondence**, utilizing features on mesh vertices to align 3D parts with 2D images. We also propose a deformation network to reshape the parts into diverse geometries. Building upon the learned part compositional neural mesh, we propose an inference pipeline capable of predicting the 3D configuration of object parts from a single image with **Render-and-Compare** method. Our experiments on 2D segmentation show that our method outperforms state-of-the-art 2D segmentation approaches with pseudo-labeling. The experiments on 3D recognition demonstrate the effectiveness of our framework on pioneering one-shot 3D part detection from a single image.

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

# A  METHOD DETAILS

## A.1  FEATURE INTERPOLATION PROCESS DETAILS

During the learning of mesh-to-mesh correspondence, we apply the feature interpolation process to compute the feature for every vertex on the meshes. Specifically, we train the Point-Net++ (Qi et al., 2017b) encoder with sampled 1024 points $\{p_{y,i}\}$ from each mesh $y$, thus obtaining the features $\{\Gamma_{y,i}\}$ for those vertices. Then the features $\{\gamma_{y,k}\}$ on all vertices $\{V_k\}$ of the mesh $y$ are computed by the weighted sum of features of neighboring sampled vertices: $\gamma_{y,k} = \frac{1}{\sum_{j \in \mathcal{N}(k)} e^{w_{kj}}} \sum_{j \in \mathcal{N}} e^{w_{kj}} \Gamma_{y,j}$, where $\mathcal{N}$ denotes the neighboring vertices and $w_{kj}$ denotes the reciprocal of euclidean distance between vertex $k$ and vertex $j$.

## A.2  DEFORMATION NETWORK DETAILS

To train the deformation network, for each category, we use the annotated mesh as the template mesh, base on which the network predicts the 3D offsets of vertices given a shape latent. The length of the shape latent is set to 8 in our implementation. We use 8 meshes in the category to train the network, which should deform the mesh into those meshes given the corresponding one-hot latent vectors. During inference, the deformed mesh could be seen as an interpolation among the 8 meshes.

## A.3  RENDERING VISIBILITY DETAILS

During Mesh-to-image correspondence training, we render the depth map $\mathbf{D} = Render(\mathfrak{N}_y, \Omega)$ and the vertex-to-camera distance $\mathbf{d}_k = \|Q - V_k\|_2$. Then the vertex visibility is computed as

$$o_k = \begin{cases} 0, \|\mathbf{D}_{p_k} - \mathbf{d}_k\|_2 > \tau_r \\ 1, \|\mathbf{D}_{p_k} - \mathbf{d}_k\|_2 \leq \tau_r \end{cases}, \tag{7}$$

where $\tau_r$ is a preset threshold.

# B  EXPERIMENT DETAILS

## B.1  DATASET DETAILS

Table 6: Number of models, images, and parts in VehiclePart3D

|  | Police Car | Airliner | Bicycle | Jeep | Minibus |
|---|---|---|---|---|---|
| CAD Model | 11 | 40 | 11 | 11 | 11 |
| Synthetic Image | 1000 | 2000 | 1000 | 1000 | 1000 |
| Real Image | 69 | 98 | 45 | 41 | 25 |
| Part Number | 4 | 7 | 4 | 7 | 7 |

Table 7: Number of images, and parts in PartImageNet

|  | Car | Aeroplane | Bicycle | Boat |
|---|---|---|---|---|
| Real Image | 127 | 81 | 70 | 64 |
| Part Number | 3 | 5 | 4 | 2 |

Table 8: Number of images, and parts in UDA-Part

|  | Car | Plane | Bicycle |
|---|---|---|---|
| Real Images | 40 | 16 | 40 |
| Part Number | 9 | 6 | 4 |

Table 9: Part Definitions for Evaluation in Vehicle3D

| Category | Part Definitions |
|---|---|
| Police Car | wheels, doors, back trunk, body |
| Airliner | left engine, right engine, fuselage, horizontal stabilizer, left wing, vertical stabilizer, right wing |
| Bicycle | wheels, body, handle bar, saddle |
| Jeep | front left wheel, front right wheel, back left wheel, back right wheel, doors, back trunk, body |
| Minibus | front left wheel, front right wheel, back left wheel, back right wheel, left door, right door, body |

Table 10: Part Definitions for Evaluation in UDA-Part

| Category | Part Definitions |
|---|---|
| Car | front left wheel, front right wheel, back left wheel, back right wheel, left door, right door, body, left mirror, right mirror |
| Plane | body, left engine, right engine, left wing, right wing, tail |
| Bicycle | front wheel, back wheel, body, saddle |

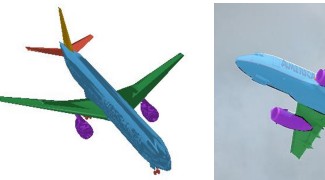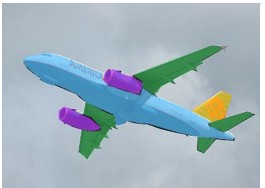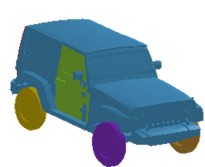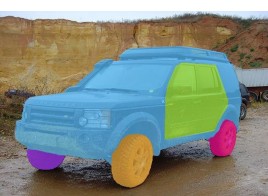

Figure 8: Example of VehiclePart3D dataset. We annotate object parts on both 3D meshes and 2D real images, making the part definition aligned between 3D and 2D.

The number of 3D CAD models, 2D images, and defined parts in each dataset are listed in Table 6, Table 7, and Table 8.

We use two different training set to validate the effectiveness of our framework: 3D CAD models from VehiclePart3D and 3D models from DST3D. For each category in VehiclePart3D (Police Car, Airliner, Bicycle, Jeep, Minibus), we use 2/3 of the CAD models and corresponding 2D images generated by diffusion models (Ma et al., 2023) for training the Mesh-to-Image correspondence. For each category in DST3D (Car, Aeroplane, Bicycle, Boat), we use 3/4 of the CAD models and generated images for training.

For the testing set, the rest of CAD models and corresponding images in VehiclePart3D are used for synthetic evaluation (including 3D detection and 2D segmentation). Due to computational limitation, we only use 25 images for each instance during testing in the synthetic evaluation. The real images in VehiclePart3D are used for 2D segmentation evaluation as shown in Table 1. The real images in PartImageNet are used for 2D segmentation evaluation as shown in Table 2. The real images in UDA-Part are used for 2D segmentation evaluation as shown in Table 3.

The specific part definitions in Vehicle3D and UDA-Part are shown in Table 9 and 10.

## B.2 VEHICLEPART3D ANNOTATION DETAILS

As shown in Figure 8, for each object category, we first define 4 to 7 semantic object parts, the parts are mainly defined based on functionality and compositionality. We annotate object parts on 3D CAD models obtained from ShapeNet (Chang et al., 2015) by manually assigning the part label to each mesh vertex. Specifically, we first label the mesh vertices into groups representing different parts in Blender (Community, 2018), and then convert the annotation into json files. For the real images, we select 279 untruncated images from the test and validation set of ImageNet (Krizhevsky et al., 2012). We then manually annotate the segmentation mask for each object part following the

Figure 9: 2D segmentation and 3D parts predicted by Ours on VehiclePart3D.

part definition on 3D CAD models leveraging SAM (Kirillov et al., 2023) for object masks with additional human annotations in other parts except body/fuselage. Human checks are conducted to ensure both the 3D and 2D annotation quality. The whole dataset is included in the supplementary material.

## B.3 TRAINING DETAILS

During training, we randomly choose 1024 vertices from each mesh for mesh-to-mesh correspondence finding and mesh-to-image alignment. The visibility threshold $\tau_r$ is set to 0.003. Neighboring vertices $\mathcal{N}_k$ of vertex $V_k$ is defined as vertices that have distances smaller than 0.03 with $V_k$. We use Adam as our optimizer with initial learning rate as $10^{-4}$. We trained for 800 epochs in each category. During inference, The consistency weight $w_{consistency}$ is set to 0.5. We use Adam optimizer with initial learning rate 0.01. In the experiment, the scales of objects are normalized to one unit. It takes about $40s$ to predict the 3D parts from an image on a single NVIDIA RTX TITAN GPU.

## B.4 NETWORK ARCHITECTURE DETAILS

We use Resnet 50 as our backbone to extract features from 2D images. The mesh features are stored in a memory bank with size of $1029 \times 128$, where 128 is the channel size of features and 1029 includes 1024 features corresponding to 1024 vertices and 5 features representing the background.

## B.5 BASELINE DETAILS

For the self-labeling setting in our baseline, we follow the online self-training framework in DAFormer (Hoyer et al., 2022). We disable the Thing-Class ImageNet Feature Distance (FD) in it. It is a regularization technique that uses ImageNet features which are trained from objects to provide guidance to segment object classes, which is inappropriate for segmenting semantics parts of object. In addition, we also disable the Rare Class Sampling (RCS) to correspond with our method which does not include sampling strategies.

## B.6 COMPUTATIONAL RESOURCE

Our training process takes about 16 hours on four NVIDIA RTX TITAN GPUs for each category. It takes about 40s on a NVIDIA RTX TITAN GPU for inferring both 3D parts and 2D segmentation on one image.

## C MORE VISUALIZATIONS

Figure 9 shows more 2D segmentation and 3D parts prediction of ours on VehiclePart3D.

As shown in Figure 10, SegFormer could not distinguish background with objects and fails to tell the spacial relation between different parts (e.g., front wheel and back wheel are predicted into one

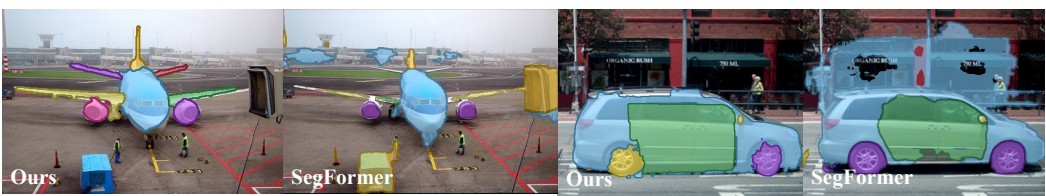

Figure 10: Predicted segmentation of Ours and SegFormer on UDA-Part.

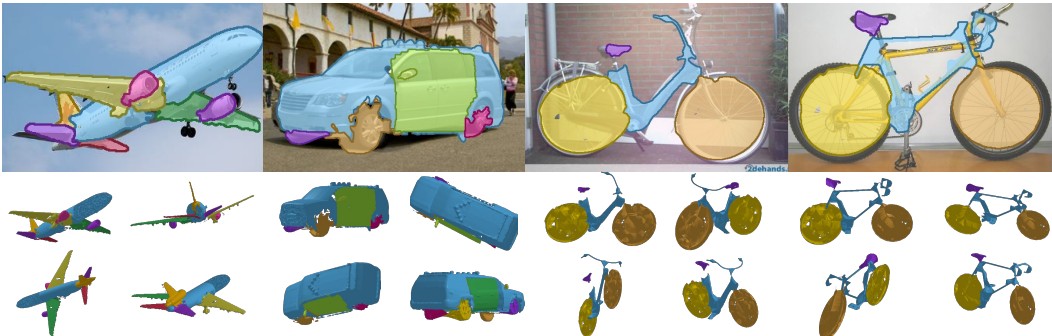

Figure 11: Predicted 2D segmentation and 3D parts of Ours on UDA-Part.

part), whereas our framework could locate parts with precise spacial information and clear object boundary.

Figure 11 shows more visualization of prediction on UDA-Part, including Car, Plane and Bicycle. Notably, the two selected bicycles demonstrate that our method could effectively choose parts from the part library regarding to the object in the image. Combined with Figure 9, It shows that our framework could apply to diverse part definitions.

## D   LIMITATIONS

Despite the fact that our framework applies many strategies to overcome the shape variance across different objects in the same category, some significant shape differences could still hinder the application of our method. For example, the shape difference between bicycle and tandem bicycle could influence the performance of our method in the bicycle category, especially considering that the part definitions on those objects are different. However, we argue that this issue could be solved by classifying subcategories or train our part deformation network on a larger scale to improve the robustness to shape variance.

