# OpenReview forum: "Part321: Recognizing 3D Object Parts from a 2D Image Using 1-Shot Annotations"
_ICLR.cc/2025/Conference — ICLR 2025 Conference Withdrawn Submission_

### Official Review · Reviewer_8Chn · 2024-11-03

**Soundness:** 3
**Presentation:** 2
**Contribution:** 2
**Rating:** 3
**Confidence:** 3

**Summary:**

The paper proposes a category-level pipeline aimed at estimating corresponding 3D parts from a single image. The key aspect of the problem formulation is that it only requires part annotation on a single mesh per category, essentially achieving one-shot annotation. The main insight of the paper lies in learning a category-level neural template mesh and performing pose estimation and 3D part reconstruction through 2D-3D feature matching between the 2D image and the neural template mesh. To accomplish this, the paper adopts a multi-stage training approach to separately learn mesh-mesh matching, mesh-image matching, and category-level dense features. During inference, optimization of object pose, part shape, and scale is achieved through a render-and-compare approach, enabling the recovery of 3D parts from images.

The primary contributions of the paper include the integration of existing techniques to achieve accurate 3D parts reconstruction; utilizing a multi-stage training process to enable mesh-mesh and mesh-image matching through unsupervised learning and pseudo-label construction without requiring annotations; and designing a render-and-compare optimization procedure and a mesh deformation network in the inference stage to achieve accurate part recovery.

**Strengths:**

- This paper achieves category-level monocular 3D part segmentation with only one-shot part annotation by integrating existing techniques. To establish correspondence between any input image and a single CAD model with part annotations, it utilizes unsupervised learning for mesh-mesh matching and pseudo-label construction for learning mesh-image matching.
- By leveraging unsupervised training for mesh-mesh matching, the mesh deformation network is trained, allowing for improved alignment during the render-and-compare optimization process through shape latent optimization.

**Weaknesses:**

- The paper lacks some key experiments that would help assess the significance of the method:
    - Necessity of render-and-compare: What would be the difference if, during inference, the approach did not rely on render-and-compare optimization but instead directly relied on feature matching between the image and the neural template mesh to obtain 2D part segmentation? How would this approach compare to the inference pipeline proposed in the paper?
    - Alternative approaches based on 2D-3D matching (or canonical mapping): Several methods (e.g., [1, 2, 3]) could also achieve part estimation for images in the same category from a single annotated mesh by label transfer through matching. One of these methods should be considered as a baseline for 2D part estimation.
- The proposed method achieves part segmentation with one-shot annotation by integrating existing methods; however, the problem formulation itself lacks significance. Technically, it overlaps heavily with category-level object pose estimation. For instance, [4] already demonstrates the effectiveness of combining NeMo and 3D-DST for object pose estimation, while [5] applies deformable template meshes to object pose estimation. Simply adapting these techniques for part annotation does not appear sufficiently significant.
- The inference process of the method relies on optimizing multiple parameters at the part level (pose, scale, shape latent), and shape latent training uses only 8 instances per category for instance-level training. This raises concerns about the robustness of the optimization process.


[1] Canonical Surface Mapping via Geometric Cycle Consistency

[2] SELF-SUPERVISED GEOMETRIC CORRESPONDENCE FOR CATEGORY-LEVEL 6D OBJECT POSE ESTIMATION IN THE WILD

[3] To The Point: Correspondence-driven monocular 3D category reconstruction

[4] GENERATING IMAGES WITH 3D ANNOTATIONS USING DIFFUSION MODELS

[5] Neural Textured Deformable Meshes for Robust Analysis-by-Synthesis

**Questions:**

- The explanation of the deformation network in the paper is unclear. According to the description in Sec-3.1, it appears that part-level latents are used, but in Sec-3.2, the shape latent is described as an instance-wise one-hot vector, yet it is referred to as a part deformation network. In the inference process in Sec-3.4 and in Figure 3, both an instance-level one-hot latent and part-level continuous latents are involved. What is the relationship between the instance-wise latent and part-level latent? If an instance-wise one-hot latent is used during training but treated as a continuous one for optimization per part during inference, additional experiments are needed to demonstrate the deformation results corresponding to latent interpolation, thereby verifying its feasibility.
- Figure 3 uses “Part Features,” whereas Sec-3.4 describes “pixel-wise features.” During optimization, does the rendered feature enforce the same feature constraint for each part?

---

### Official Review · Reviewer_uktx · 2024-11-04

**Soundness:** 3
**Presentation:** 2
**Contribution:** 3
**Rating:** 5
**Confidence:** 4

**Summary:**

This work presents Part321, a 2D/3D representation learning technique that allows for 2D part segmentation and 3D part shape and pose estimation given a single 3D labeled mesh at inference time.

To learn this for an object category, the method needs an aligned 3D mesh dataset rendered into images with known cameras. A mesh-to-mesh correspondence is learned through learning to a per-vertex deformation transform across meshes, and a mesh-to-image correspondence is learned by image-to-mesh feature contrastive loss using known 2D pixel to 3D vertex correspondences.

At inference time, given a single labeled 3D mesh, an optimization procedure can be used to deform the label mesh into a mesh close to the object shown in the 2D test image, and optimize the pose of the complete object and object parts. At the end of this procedure the result is a one-shot 3D and 2D part estimate. The Part321 outperforms with 2D segmentation models trained on the same labeled limited data that Part321 uses, including some domain adaptation techniques to boost their performance. Part321 is also quantitatively evaluated at 3D part shape and pose estimation, but without baseline comparisons as there are no prior methods.

**Strengths:**

### Originality
- The motivation behind Part321 for one-shot part segmentation is sound: learning how to consistently encode points on the surface of an object when represented with 3D vertices and image pixels can only be done if the underlying encoders reason about object parts. This design directly allows for test-time optimization with one labeled 3D mesh for high accuracy 2D and 3D part estimation. Our knowledge of 3D geometry is applied well to try and solve problem of part understanding.
- There are no prior works that use both 3D correspondence and 2D to 3D correspondence and contrastive training
### Quality
- There are thorough ablation studies of the proposed model features
- Qualitative and quantitative results show improved performance over 2D segmentation baselines, and reasonable quantitative and qualitative results on estimating 3D shape and pose of object parts
### Clarity
- The paper is reasonably well written and easy to follow, there are some minor notes in the weaknesses.
### Significance
- This work identifies how paired 2D and 3D synthetic data can be used for one shot 2D and 3D part understanding for a single object category at a time. It sets the stage for future work in part understanding to simultaneously investigate inference in both 3D and 2D.

**Weaknesses:**

- The proposed approach requires 3D datasets of aligned meshes for each category of interest. Further, a separate model needs to be trained for each category. This limits the scalability and applications as it requires both aligned 3D data and many models. It also makes it difficult to extend to deformable or articulated objects.
- Evaluations are only done on four categories (car, aeroplane, bicycle and boat). Is it possible to use datasets like [PACO (CVPR23)](https://openaccess.thecvf.com/content/CVPR2023/html/Ramanathan_PACO_Parts_and_Attributes_of_Common_Objects_CVPR_2023_paper.html) or [PartNet-E](https://openaccess.thecvf.com/content/CVPR2023/html/Liu_PartSLIP_Low-Shot_Part_Segmentation_for_3D_Point_Clouds_via_Pretrained_CVPR_2023_paper.html)? It's difficult to gauge the significance or true capability of Part321 unless larger and more complex datasets are used.
- Baselines are 2D segmenters trained on small labeled datasets which is likely suboptimal for those methods. It is important to include comparisons to off-the-shelf methods such as [Matcher (ICLR24)](https://github.com/aim-uofa/Matcher) which are off the-shelf-techniques that can be directly applied to 2D part segmentation. This is not a one-to-one comparison, but is an important datapoint to understand the significance of what Part321 has achieved in an absolute sense.


Minor
- L151 - "specially accuracy" typo
- L248 - $\theta$ is not defined as a variable

**Questions:**

The main questions relate to the weaknesses above: it would be great if the authors can discuss based the quality of which I'd be happy to increase my rating.
- How such an approach can scale to many diverse categories, including possibly articulated and deformable objects?
- Is there a reason why larger scale evaluations were not conducted?
- It would be great to have a comparison with an off-the-shelf zero-shot method like Matcher, which would allow the reader to better understand the significance of what's being achieved by Part321.

---

### Official Review · Reviewer_suZT · 2024-11-04

**Soundness:** 3
**Presentation:** 1
**Contribution:** 2
**Rating:** 5
**Confidence:** 4

**Summary:**

The paper introduces a novel approach for one-shot 2D part segmentation, leveraging part-level 3D shape reconstruction from a single part-annotated 3D CAD model as a template. This approach optimizes 3D part shapes to align with target images at the feature level, achieving per-part 2D segmentation through projection. The method uniquely combines 3D shape reconstruction with 2D segmentation, learning mesh-to-mesh and mesh-to-pixel correspondences in a self-supervised manner. Evaluated on three real-world datasets, it consistently outperforms traditional 2D segmentation baselines without domain adaptation (DA) and performs similarly or better when DA is applied.

**Strengths:**

- The proposed approach is a novel direction for one-shot 2D part segmentation, combining 3D part-level shape reconstruction.
- The paper addresses the challenging task of generalizing 3D-to-2D correspondence and shape reconstruction in camera space in an unsupervised manner.
- The proposed method outperforms the baseline 2D semantic segmentation models with the additional domain adaptation technique.

**Weaknesses:**

## Major
- The notation and writing are confusing and difficult to follow.
  - L193: Should \( z_k \) be \( z_y \)? Is it a per-mesh shape latent feature rather than per-vertex? Additionally, Fig. 3 and L295 suggest that each part shape may have an independent shape latent.
  - L188 and L234: Is \( U_k \) in L188 the same as \( \theta_j \) in L234?
  - The notation lacks clarity for the learnable parameter used for the image encoder \( \psi \), creating confusion, as it seems the vertex deformation network \( \phi \) is not learnable.
  - L228: \( \phi \) is reused as an image encoder, although it was previously defined as the mesh deformation network in L193.
  - L243: Notation for \( j \) and \( i \) is missing.
  - L238: Why is the camera pose represented as \( \mathbb{R}^3 \)? Is it only for translation? Why is rotation ignored and not represented as \( SE(3) \)?
  - In Eq. 3, what is \( \mathcal{y} \)?
  - L251: The background term is not defined. Is it Eq. 3 of [Bai et al., 2023]?
  - L269: Does \( W \) represent trained weights?
  - Figure 2: The figure and notation in the manuscript do not match, making the pipeline hard to understand. The trainable modules, such as the mesh deformation network (L193), vertex feature encoder (L205), and image encoder (L228), should be clearly indicated.

- A comparison with foundation-based models would be preferable. Foundation models exhibit part-level segmentation capabilities in a one-shot setting (e.g., Fig. 3 of [1]). Additionally, previous work has leveraged this capability for category-level, part-level 3D shape reconstruction using only 2D images [2]. Since these approaches do not require per-part annotation nor 3D models with or without annotation, additional experiments/explanations for comparison would be beneficial to clarify the advantage of the proposed method.

## Minor
- The idea of a neural mesh with "render and compare" (learning a surface embedding for the template mesh and corresponding per-pixel feature encoder for image-aligned shape reconstruction) is similar to [3,4] but is not referenced. Although these methods require a pretrained DensePose model [3] or are purely optimization-based [4], the manuscript should at least discuss the differences from these methods.

# References
1. Liu et al. "MATCHER: Segment Anything with One Shot Using All-Purpose Feature Matching." ICLR 2024.
2. Yao et al. "Hi-LASSIE: High-Fidelity Articulated Shape and Skeleton Discovery from Sparse Image Ensemble." CVPR 2023.
3. Kulkarni et al. "Articulation-aware Canonical Surface Mapping." CVPR 2020.
4. Yang et al. "ViSER: Video-Specific Surface Embeddings for Articulated 3D Shape Reconstruction." NeurIPS 2021.

**Questions:**

The unclear parts in the manuscript and suggestions on foundation models are included in the weakness.

---

### Official Review · Reviewer_FCGe · 2024-11-04

**Soundness:** 3
**Presentation:** 3
**Contribution:** 3
**Rating:** 6
**Confidence:** 4

**Summary:**

The paper proposes a model for recognizing 3D object parts from a single RGB image by inferring the 3D configuration of each object part. The model is one-shot in that it requires annotations of parts on one 3D mesh model for each object category. The model incorporates mesh-to-mesh correspondence determination for alignment of distinct 3D mesh models within an object category and mesh-to-image correspondence for alignment of a 3D model with a 2D RGB image.

**Strengths:**

The paper tackles an important problem. The proposed methodology is technically sound and the results are encouraging when compared to the other state-of-the-art models.

**Weaknesses:**

The robustness of the proposed technique to occlusions and other image degradations has not been discussed or studied. For example, it is not clear how robust the proposed technique is to partial occlusion of an object part in an image. This would apply to both, the training phase where the correspondences are established and the testing phase where the object parts are recognized. The authors to test their proposed technique on synthetically occluded images and images with added degradations (such as noise and blur), both during training and testing phases. The extent of occlusion and image degradation should be systematically varied in terms of extent and intensity respectively and the performance of the proposed technique quantified and noted accordingly.

**Questions:**

How robust/sensitive is the proposed model to occlusions and image degradations? How sensitive are the final results to the choice of the various loss functions discussed in the paper? To answer the above the authors are advised to perform a sensitivity analysis with varying levels of image occlusions and image degradations (Gaussian noise, motion blur etc.). A formal ablation study based on the loss function type and loss function weights would be very desirable.

---

### Official Review · Reviewer_UVT2 · 2024-11-05

**Soundness:** 2
**Presentation:** 2
**Contribution:** 2
**Rating:** 5
**Confidence:** 3

**Summary:**

This paper introduces a model for identifying 3D object parts from 2D images using a single annotated 3D mesh per object category. It builds a 3D feature bank to align 3D parts with 2D images by learning mesh-to-mesh and mesh-to-image correspondences. During inference, the model optimizes part configurations to match image cues, ensuring geometric consistency. Tests on datasets like VehiclePart3D and PartImageNet show the model's superior performance in both 2D and 3D part recognition tasks.

**Strengths:**

+ Propose an interesting setting that recognize 3D parts of object from single image.

+ Design two different correspondences to bridge 2D images and 3D meshes.

**Weaknesses:**

- The authors state in line 108 that a single 3D annotation is used for each class, yet they mention in line 323 that 'During training, we use 30 CAD models … for each category.' This creates ambiguity around the training setup. Does this imply that out of the 30 CAD models, only one is annotated, or are additional annotations being used in some form?

- The evaluation metric used in this paper appears misaligned with the task objectives. While the primary focus of the paper is on 3D part recognition, instead of using 3D segmentation metric, the authors rely on detection metrics for evaluation, which may not effectively capture segmentation performance.

- The experiments on 2D part segmentation in Table 1 are limited in scope. The paper only compares against earlier work, without including recent methods  (e.g., [1])  specifically designed for 2D part segmentation.

- In the 3D part segmentation experiments, the authors do not include comparisons with any baseline methods (e.g., [2,3]) that address similar tasks. While training settings may differ slightly, such comparisons are essential to contextualize the performance and validate the effectiveness of the proposed approach.

- The experiments appear limited to only a few categories (e.g., Car, Aeroplane, Bicycle, Boat). I believe results on more categories should be included to show the effectiveness of the proposed method.


-------
[1] Learning Part Segmentation through Unsupervised Domain Adaptation from Synthetic Vehicles, CVPR'22

[2] PartSLIP: Low-Shot Part Segmentation for 3D Point Clouds via Pretrained Image-Language Models, CVPR'23

[3] 3x2: 3D Object Part Segmentation by 2D Semantic Correspondences, ECCV'24

**Questions:**

See Weakness.

---

### Note · Authors · 2024-11-15

**Comment:**

Thanks all the reviewer for the feedback. We will revise the paper based on the comment.

**Withdrawal Confirmation:**

I have read and agree with the venue's withdrawal policy on behalf of myself and my co-authors.